# Pro-Angiogenic and Osteogenic Effects of Adipose Tissue-Derived Pericytes Synergistically Enhanced by Nel-like Protein-1

**DOI:** 10.3390/cells10092244

**Published:** 2021-08-30

**Authors:** Hyun-Ju An, Kyung Rae Ko, Minjung Baek, Yoonhui Jeong, Hyeon Hae Lee, Hyungkyung Kim, Do Kyung Kim, So-Young Lee, Soonchul Lee

**Affiliations:** 1Department of Orthopaedic Surgery, CHA Bundang Medical Center, CHA University School of Medicine, 59 Yatap-Ro, Seongnam-si 13496, Gyeonggi-do, Korea; yks486ahj@naver.com (H.-J.A.); eclsa79@gmail.com (M.B.); jeongyunhui92@gmail.com (Y.J.); aotcnlsl@gmail.com (H.H.L.); 2Department of Orthopaedic Surgery, Samsung Medical Center, Sungkyunkwan University School of Medicine, 81 Irwon-Ro, Gangnam-gu, Seoul 06351, Korea; krmd.ko@gmail.com; 3Department of Pathology, Kyung Hee University Hospital at Gangdong, Kyung Hee University, College of Medicine, 892 Dongnam-Ro, Gangdong-gu, Seoul 05278, Korea; hyungkyungjkim@gmail.com; 4CHA Graduate School of Medicine, 120 Hyeryong-Ro, Pocheon-si 11160, Gyeonggi-do, Korea; dokim1018@naver.com; 5Department of Internal Medicine, CHA Bundang Medical Center, CHA University School of Medicine, 59 Yatap-Ro, Seongnam-si 13496, Gyeonggi-do, Korea; ysy0119@cha.ac.kr

**Keywords:** pericytes, *Nel*-like protein-1, angiogenesis, bone regeneration, bone formation, osteonecrosis

## Abstract

An important objective of vascularized tissue regeneration is to develop agents for osteonecrosis. We aimed to identify the pro-angiogenic and osteogenic efficacy of adipose tissue-derived (AD) pericytes combined with Nel-like protein-1 (NELL-1) to investigate the therapeutic effects on osteonecrosis. Tube formation and cell migration were assessed to determine the pro-angiogenic efficacy. Vessel formation was evaluated in vivo using the chorioallantoic membrane assay. A mouse model with a 2.5 mm necrotic bone fragment in the femoral shaft was used as a substitute for osteonecrosis in humans. Bone formation was assessed radiographically (plain radiographs, three-dimensional images, and quantitative analyses), and histomorphometric analyses were performed. To identify factors related to the effects of NELL-1, analysis using microarrays, qRT-PCR, and Western blotting was performed. The results for pro-angiogenic efficacy evaluation identified synergistic effects of pericytes and NELL-1 on tube formation, cell migration, and vessel formation. For osteogenic efficacy analysis, the mouse model for osteonecrosis was treated in combination with pericytes and NELL-1, and the results showed maximum bone formation using radiographic images and quantitative analyses, compared with other treatment groups and showed robust bone and vessel formation using histomorphometric analysis. We identified an association between FGF2 and the effects of NELL-1 using array-based analysis. Thus, combinatorial therapy using AD pericytes and NELL-1 may have potential as a novel treatment for osteonecrosis.

## 1. Introduction

Angiogenesis is a vital process for normal tissue development and supports skeletal growth. It is also associated with a variety of pathological conditions [1,2]. Pathological angiogenesis is a hallmark of various ischemic conditions [3]. Therefore, restoring vascular homeostasis and therapeutic angiogenesis have great potential for the treatment of ischemic conditions. The pursuit of a safe and efficacious approach to restoring angiogenesis is a field of ongoing research [2].

Avascular necrosis is defined as the cellular death of bone caused by interruption of normal blood flow. It most frequently affects the femoral head (i.e., osteonecrosis of the femoral head, ONFH). The affected limb becomes progressively more painful and debilitated with progressive collapse of subchondral bone and development of osteoarthritis. Total hip arthroplasty is regarded as a reliable surgical procedure in the treatment of advanced ONFH, but the lifetime risk of revision is a concern, especially in young patients [4]. Core decompression is one of the most commonly performed surgical procedures for early-stage ONFH. However, ideal indications of core decompression have been debated [5], and recent studies have focused on its use combined with regenerative techniques rather than its use alone in terms of improved results [6]. To overcome the limitations of current treatments, it is highly desirable to develop agents to enhance angiogenesis and bone formation.

As a regenerative therapeutic option in the field of orthopedic surgery, significant effects of human perivascular stem cells combined with Nel-like protein-1 (NELL-1) in angiogenesis and osteogenesis have been demonstrated using a mouse model of ectopic bone formation and [7] a rat model of osteoporosis [8]. Considering the need for novel therapeutic options in ONFH, we focused on their synergistic effects shown in the previous animal models [7,8].

Perivascular stem cells are found in all vascularized tissue; therefore, they are abundant and easily harvestable [9]. As they can be used after sorting, bypassing culture expansion, the extra time and risks related to ex vivo expansion can be eliminated [7]. Perivascular stem cells consist of (1) pericytes, which line micro-vessels, and (2) adventitial cells found in the tunica adventitia of large arteries and veins [9,10,11,12]. Pericytes have critical roles in the regulation of endothelial cell function and angiogenesis [10,12,13,14]. During angiogenesis, pericytes participate in recruitment, extracellular matrix modulation, paracrine signaling, and direct interactions with endothelial cells [15]. Study results indicate that pericytes participate through direct cell contact and communication. Their main functions include angiogenesis stimulation, regulation of blood vessel diameter [15], and maintenance of vascular permeability and integrity [16]. Because of their effects on endothelial cells, the study of pericytes is a research area of increasing interest that includes evaluation as potential targets of pro-angiogenic therapies [17]. While their presence in microcirculation has long been documented, their functional roles and importance have been largely under-investigated.

As mentioned above, NELL-1 is a potential additive to enhance angiogenesis and bone formation [7,8]. It was first identified as an osteoinductive protein via its overexpression in cranial sutures [18]. It has robust bone formation effects in the axial skeleton and the extremities in in vivo settings [19,20,21,22,23,24]. Further, NELL-1 can stimulate the proliferation of perivascular stem cells [25] and has anti-osteoclastic effects [26].

On this background, we conducted a study on the synergistic effects of pericytes combined with NELL-1. Our examination of the potential for this combination as a novel therapeutic agent for osteonecrosis included the use of a mouse model with a necrotic bone fragment.

## 2. Methods

### 2.1. NELL-1 and Human Stromal Vascular Fraction (HSVF) Isolation from Human Adipose Tissue and Purification of Human Pericytes from HSVF

The NELL-1 was kindly provided by Professors Kang Ting, Chia Soo, and Xinli Zhang (UCLA, USA). Human adipose tissue was obtained from 8 patients who underwent total knee arthroplasty due to degenerative osteoarthritis (Table 1). The adipose tissue samples were stored (4 °C) until they were processed. All samples were processed within 48 h after collection. Human stromal vascular fraction (hSVF) was prepared by digesting the adipose tissue using collagenase digestion, as previously described [8]. The adipose tissue was briefly washed in an equal volume of phosphate-buffered saline (PBS). Collagenase digestion was performed using Dulbecco’s Modified Eagle’s Medium (DMEM; Sigma-Aldrich, St. Louis, MO, USA) containing 3.5% bovine serum albumin (Sigma-Aldrich) and 1 mg/mL collagenase type II for 70 min under agitation at 37 °C. The filtered solution was centrifuged to separate and exclude adipocytes. The processed hSVF was suspended in red cell lysis buffer (155 mM NH_4_Cl, 10 mM KHCO_3_, and 0.1 mM EDTA) and incubated for 10 min at room temperature. The hSVF filtrate was immediately processed for human pericyte purification.

A fluorescence-activated cell sorter (FACS) was used to purify human pericytes from isolated hSVF, as previously described [11,27]. The isolated hSVF was centrifuged, and the resulting pellet was incubated (4 °C for 15 min in the dark) with conjugated 1:100 dilutant antibodies (anti-CD34-phycoerythrin (Dako), anti-CD45-allophycocyanin (Santa Cruz Biotechnology, Inc., Dallas, TX, USA), and anti-CD146-fluorescein isothiocyanate (AbD Serotec)). The hSVF pellet was then resuspended in PBS and 4′, 6-diamidino-2-phenylindole (Invitrogen, Carlsbad, CA, USA) and was filtered through a 70 μm cell filter for removal of nonviable cells. The solution was processed on a FACS Aria cell sorter (BD Biosciences, San Jose, CA, USA) to isolate populations of cells that constituted human pluripotent stem cells based on cell surface markers: pericytes (CD146+, CD34−, CD45−).

### 2.2. Osteogenic and Adipogenic Differentiation of Human Pericytes

The human pericytes were cultured separately in the PBS control and NELL-1 solutions to determine the differentiation potential of human pericytes and to test the biologic activity of NELL-1 protein during osteogenesis and adipogenesis. Osteogenic differentiation of the pericytes occurred over a period of 15 d. The cells were added to 24-well plates (3 × 10^4^ cells/well density) with DMEM + 10% fetal bovine serum (FBS). Within 24 h, osteogenic differentiation of the cells was induced in the PBS control and NELL-1 (800 ng/mL) treatments using osteogenic differentiation medium (DMEM + 10% FBS + 50 µg/mL ascorbic acid, and 3 mM β-glycerophosphate). The medium was changed every 3 d. Alizarin Red staining was used to assess osteogenic differentiation.

For adipogenic differentiation, human pericytes were added to 24-well plates (5 × 10^4^ cells/well density) with DMEM + 10% FBS. In 24 h, cells were induced to adipogenic differentiation in the PBS control and NELL-1 (800 ng/mL) treatments using adipogenic differentiation medium (Human MesenCult™ Adipogenic Differentiation Medium; STEMCELL TECHNOLOGIES, Catalog #05412). Adipogenic differentiation was performed over 12 d. The medium was changed every 3 d. Adipogenic differentiation was assessed at 12 d using Oil Red O staining.

### 2.3. Endothelial Cell Culture

Human umbilical vein endothelial cells (HUVECs) were purchased from Lonza and cultured in an endothelial growth medium (Lonza, Basel, Switzerland). The cells were used between passages 4 and 7 [28].

### 2.4. Viability Test

HUVECSs were seeded in 96-well plates (4 × 10^4^ cells/well) and treated with pericyte and NELL-1. A total of 2000 pericytes were added to 100 μL DMEM in each well of two 96-well plates. The cells were incubated in DMEM with 800 ng/mL NELL-1 for 72 h. Control cells were incubated with only DMEM. A water-soluble tetrazolium salt (WST) assay (Cell Counting Kit-8; Dojindo, Kumamoto, Japan) was used to measure cell proliferation. Then, WST (10 µL) was added to each well and the cultures were incubated for an additional 2 h at 5% CO_2_ and 37 °C before evaluation using spectrophotometry. Conversion of WST to formazan was measured at 450 nm. The results were normalized and were presented as a percentage of the viable cells in the control group [29,30].

### 2.5. Tube Formation Assay

A capillary tube formation assay (Matrigel, Corning, Corning, NY, USA) was used to assess the effects of human pericytes and NELL-1 on endothelial cell morphogenesis. Briefly, 4 × 10^4^ cells per well were added to Matrigel-pre-coated 96-well plates and treated with pericytes (0.2 × 10^4^ cells/well) or NELL-1 (800 ng/mL), or both. Suramin (40 mM; Sigma-Aldrich) was included as a negative control. After 12 h, cell changes were recorded using a microscope (Nikon ECLIPSE Ti2) and analyzed using the Image J software program. This program provided automated quantitative measurements of tube characteristics (e.g., number of connected tubes, tube area, and angiogenic index).

### 2.6. Wound Migration Assay

HUVECs were added at a density of 3 × 10^3^ cells per well in the ibidi Culture-Insert 2 Well in a µ-Plate 24 Well plate and allowed to grow into a confluent monolayer overnight. A fresh medium containing indicated concentrations of human pericytes, and NELL-1 was added. The Culture-Insert 2 Well was then removed with sterile tweezers. After 24 h, the cells were photographed using a JuLI™ Stage Cell History Recorder (NanoEntek, Guro-gu, Seoul, Korea). The migration rate was calculated using the JuLI™ Stage. A total of 3 measurements of 3 independent wounds were taken for each monolayer sample.

### 2.7. Chick Chorioallantoic Membrane Assay

The chick chorioallantoic membrane (CAM) assay was used to assess the effect of NELL-1 on ex vivo angiogenesis. Briefly, fertile chicken eggs were candled on embryonic day 3. A small opening was made at the top of the live eggs, and a sterilized Thermanox™ Coverslip (Nunc™) saturated with either PBS or NELL-1 was placed on the CAM. The holes were then sealed with cellophane tape. The eggs were photographed after a 72 h incubation. Image J software was used to quantify blood vessel density; the results were presented using bar diagrams [30].

### 2.8. Animals

A total of 24 8-week-old male NOD SCID mice (CHA Institute Animal Experimentation, Pangyo, Seongnam, Korea) were used for the study to prevent immune reactions to implants containing human cells. Each mouse was housed alone in a pathogen-free ventilated cage, fed a standard rodent chow diet, provided tap water *ad libitum*, and experienced 12 h light and dark cycles. The mice were cared for following the Chancellor’s Animal Research Committee for Protection of Research Subjects guidelines at the CHA medical university (IACUC190082).

### 2.9. Implant Preparation and Grouping

Recombinant human NELL-1 was purchased from Bone Biologics Inc. (UCLA, Los Angeles, LA, USA). An absorbable collagen sponge (ACS) of defined dimensions (0.5 cm × 0.5 cm × 1.0 cm, Lyoplant, 1066102; Aesculap AG, Tuttlingen, Germany) was used. Because intramuscular implantation of ACS alone has no known bone-forming effects, this carrier was chosen for its non-osteoinductive characteristics [31]. Defined concentrations of viable cells and NELL-1 in PBS suspension (20 µL) from the hSVF were applied and allowed to saturate the ACS. The cell and protein scaffold suspensions were kept on ice until implantation. The 4 treatment groups used were (a) control with PBS (b) high concentration (800 ng/mL) NELL-1, (c) pericytes, and (d) pericytes loaded with a high concentration (800 ng/mL) of NELL-1.

### 2.10. Surgical Procedure for Mouse Models of Osteonecrosis

We used a mouse model with a necrotic bone fragment to replace osteonecrosis in humans. A total of 30 mice (6 for control, 8 for NELL-group, 8 for the pericytes group, and 8 for the pericytes + NELL-1 group) were used for the experiment. All mice were prepared at the age of 3 months. Anesthesia was initiated in a 5% gaseous isoflourane-filled holding chamber and maintained with 3–4% gaseous isoflourane through a nose cone. Before making an incision, the thigh of the mouse was shaved, and the skin was prepared using an alcohol and betadine solution. Sterile ophthalmic lubricant ointment was applied to each eye, and a buprenorphine injection (0.05 mg/kg) was given via the subcutaneous route.

All surgical procedures were performed by the senior author, an experienced orthopedic surgeon. The femoral bone was exposed using an incision made on the anterolateral aspect of the thigh. A PEEK plate was located on the anterior femur. The most proximal hole of the plate was gently drilled using a 0.3 mm drill bit, and the first screw was inserted. Additional distal screws were inserted in a similar fashion. The 0.22 mm Gigli saw wire was closely placed around the bone in a mediolateral orientation and inserted in the slots of the customized jig to create a 2.5-mm bone defect. The jig was inserted on the stem of the two last screws and applied above the plate. Next, a 2.5-mm long mid-diaphyseal femoral ostectomy was performed using the Gigli saw while applying a constant steady tension. Care was taken to avoid excess movement to obtain a straight bone cut. After the ostectomy, the Gigli saw was removed, and the saw wire was cut close to the bone on one side. The jig and remaining stem of the screw were removed. The bone fragment from the ostectomy was removed and immersed in liquid nitrogen for 5 min to induce necrosis. The ACS treated with the assigned material based on the treatment group (±pericytes ± NELL-1) was put below the femoral shaft and the necrotic bone fragment was taken back to the original site. It was then wrapped using the ACS. Finally, the ACS was sutured to prevent displacement of the necrotic bone fragment, and the wound was closed (Figure 1).

### 2.11. In Vivo Plain Radiograph

In vivo bone regeneration was assessed using plain radiography images (GIX-I, Genoray, Seongnam, Korea; Ultra Light Portable X-ray, Nanoray, Deagu, Korea) of the femur under the conditions of 70 kV/20 mA, 0.06 s; 20 lines per mm spatial resolution. Standard lateral digital radiographs of the femur were taken immediately after surgery and 4 weeks later under volatile anesthesia [32].

### 2.12. Ex Vivo Micro Computed Tomography

The mice were euthanized using CO_2_ asphyxiation 4 weeks after surgery. High-resolution micro-computed tomography (CT) scanning of each sample was performed (Bruker microCT Skyscan 1173). Each piece of femur bone was in a polyethylene tube filled with alcohol (75 volume percent) during scanning. The radiographic projections were acquired at 130 kV and 60 uA with a fixed exposure time of 500 ms, an A1 1.0 mm filter, and a 6.04 um pixel size. Four frames were averaged for each rotation increment of 0.9. Three-dimensional images with an average voxel size of 13 mm were reconstructed using a Hamming-filtered back-projection and the manufacturer’s reconstruction software (NRecon; Skyscan, Aartselaar, Belgium). Bone mineral density (BMD) of the femur was measured using the Bruker microCT with a phantom. The analyses were performed in the same manner for each mouse, with a volume of interest corresponding to the respective defect. The number of united cortices in 2 orthogonal reconstructed views was recorded for the qualitative analysis. Bone union was defined as a union of 4 out of 4 cortices. Resident software (CTAn; Skyscan, Aartselaar, Belgium) was used to obtain the BV/TV and BMD for quantitative analysis of bone formation within a region of interest. A lower gray threshold of 45 grayscale indices (attenuation coefficient of 0.035) and an upper gray threshold of 240 grayscale indices (attenuation coefficient of 0.186) were used for each mouse [33].

### 2.13. Histology and Histomorphometric Analysis

The animals were euthanized 4 weeks after surgery. Histologic specimens were fixed in 4% paraformaldehyde at 4 °C for 1 d, followed by decalcification (Calci-Clear™ Rapid Decalcifying Solution, HS-105; National Diagnostics) for 3 h at room temperature with gentle mechanical stirring. The specimens were then dehydrated and embedded in paraffin. The tissue blocks were sectioned to 3 mm thicknesses along longitudinal planes (Leica RM2235 microtome; Leica Microsystems GmbH, Wetzlar, Germany). All sections were stained with hematoxylin and eosin and alcian blue staining.

Immunohistochemistry (IHC) was performed using a ready-to-use IHC/ICC kit (BioVision, Inc.) according to the manufacturer’s protocol. Briefly, slides were deparaffinized, rehydrated, and microwaved in citrate buffer (cat. no. ab93678; Abcam) for antigen retrieval. The slides were incubated in 3% H2O2 at room temperature for 30 min to quench endogenous peroxi- dase activity, and then blocked in blocking buffer (BioVision, Inc.), followed by incubation with CD31 (Cell signaling, cat. no. #77699, 1:100), or VEGF (Abcam, cat. no. MA5-13182,1:100) antibodies at room temperature for 30 min. After incubation with HRP-anti-mouse or -rabbit IgG polymer at room temperature, the sections were treated with 3,3′-diaminobenzidine at room temperature for 10 min, followed by counterstaining. Images were captured with an Eclips-TS2 microscope.

### 2.14. Microarray

Microarrays were used to measure expression levels of genes related to NELL-1 activity. NELL-1 treated pericyte samples were used. Briefly, total RNA was isolated using RNeasy columns (Qiagen, Valencia, CA, USA) according to the manufacturer’s protocol. The RNA samples were quantified after processing with DNase digestion and clean-up procedures. An Ambion Illumina RNA amplification kit (Ambion, Austin, TX, USA) was used to amplify and purify the total RNA. Total RNA (550 ng) from each sample was converted to double-strand cDNA. Using a T7 oligo (dT) primer, amplified RNA (cRNA) was generated from the double-stranded cDNA template using an in vitro transcription reaction and purified using the Affymetrix sample clean-up module. An ND-1000 Spectrophotometer (NanoDrop, Wilmington, DE, USA) was used to quantify cDNA after purification. Uracil-DNA glycosylase and apurinic/apyrimidinic endonuclease and restriction endonucleases were used to fragment the cDNA. It was end-labeled using a terminal transferase reaction incorporating a biotinylated dideoxynucleotide. Fragmented end-labeled cDNA was hybridized to the GeneChip Human Gene 2.0 ST arrays manual (Affymetrix, Santa Clara, CA, USA). After hybridization, the chips were stained and washed (GeneChip Fluidics Station 450; Affymetrix) and scanned (GeneChip Array scanner 3000 G7; Affymetrix) by Macrogen Ltd. (Seoul, South Korea). Affymetrix^®^ GeneChip™ Command Console software was used to compute signal values.

### 2.15. RNA Isolation and QRT-PCR Analysis

Total RNA was extracted from cells using Trizol reagent (Invitrogen), according to the manufacturers’ instruction. One micrograms of total RNA were used to determine the expression of mRNAs using AMPIGENE^®^ qPCR Green Mix (Enzo Biochem, Inc., New York, NY, USA) and iCycler real-time PCR detection system (Bio-Rad, CA) according to the manufacturers’ instruction. The sequences of the primers were as follows: FGF2, 5′-AGAAGAGCGACCCTCACATCA-3′ (forward) and 5′-CGGTTAGCACACACTCCTTTG-3′ (reverse); IL-6, 5′-ACTCACCTCTTCAGAACGAATTG-3′ (forward) and 5′-CCATCTTTGGAAGGTTCAGGTTG-3′ (reverse); TGFB2, 5′-CAGCACACTCGATATGGACCA-3′ (forward) and 5′-CCTCGGGCTCAGGATAGTCT-3′ (reverse); VEGFA, 5′-AGGGCAGAATCATCACGAAGT-3′ (forward) and 5′-AGGGTCTCGATTGGATGGCA-3′ (reverse); β-actin, 5′-ACCGAGCGCGGCTACAG-3′ (forward) and 5′-CTTAATGTCACGCACGATTTCC-3′. β-actin was used for the normalization of mRNA.

### 2.16. Western Blot

Protein extraction buffer (Pro-Prep, iNtRON Biotechnology, Gyeonggi-do, Korea) was used to lyse cells for Western blot. After centrifugation (4 °C, 13,000 rpm for 15 min), protein content of lysed cells was assessed (Bradford assay). Equal total protein amounts were run on 12% SDS polyacrylamide gels and transferred to a nitrocellulose membrane. Membranes were blocked using 5% non-fat milk powder at room temperature and then incubated with primary antibodies in tris-buffered saline-tween 20 (TBS-T) overnight at 4 °C. The primary antibodies against FGF2 (cat. no. sc-271930, 1:1000), GAPDH (cat.no. sc-47724, 1:3000) were purchased from Santa Cruz Biotechnology, Inc. Antibodies against phospho-PLCγ (cat.no. #8713, 1:1000), PLCγ (cat.no #5690, 1:1000), phospho-AKT (cat.no. #4060, 1:1000), AKT (cat.no. #4691, 1:1000), phospho-ERK (cat.no. #4370, 1:1000), ERK (cat.no #4695, 1:1000), phospho-eNOS (cat.no. #9570, 1:1000), eNOS (cat.no. #32027, 1:1000) were purchased from Cell Signaling Technology, Inc.

The membranes were then washed in TBS-T and incubated with 1:5000 goat anti-mouse IgG (Santa Cruz Biotechnology) secondary antibodies for 1 h at room temperature. An enhanced luminol-based chemiluminescence detection kit (Bio-Rad Laboratories, Hercules, CA, USA) was used to visualize the resulting bands. Protein quantification was performed using densitometric digital analysis of the protein bands (ChemiDoc™ XRS+ with Image Lab™ Software ver. 6.0; Bio-Rad Laboratories). The membrane was re-probed with GAPDH (Santa Cruz Biotechnology) to confirm equal loading. After each sample was run at least 3 times on Western blot analysis, densitometry analysis was performed for each of the 3 bands.

### 2.17. Statistical Analysis

All data were presented as the mean ± SEM. All statistical analyses were performed using SPSS 17.0 (SPSS, Inc.) and GraphPad Prism software 5 (GraphPad Software, Inc.). Student’s t-test was used to compare the difference between 2 groups, and one-way ANOVA with post hoc Tukey’s test was performed to compare the differences among more than 2 groups.

## 3. Results

### 3.1. Pericytes and NELL-1 Promote Angiogenesis and Migration of HUVECs In Vitro and during Embryonic Vascularization In Vivo

To assess whether pericytes and NELL-1 affected angiogenesis of HUVECs in vitro, the HUVECs were treated with pericytes alone (pericyte group; PG), NELL-1 alone (NELL-1 group: NG) or pericytes and NELL-1 (pericytes + NELL-1 group; PNG). The results indicated that the PG had significantly greater numbers of tubes and branch points than the control group that had HUVECs alone. The presence of pericytes increased tube formation, and this increase intensified with the addition of NELL-1. Further, the addition of NELL-1 induced greater numbers of tubes and branch points than in the control group. In particular, the number of branch points in the PNG was greater than in the PG (Figure 2A). In addition, it was confirmed that the proliferation of HUVECs was slightly increased in the PNG group compared to the control group (Figure 2B). Because endothelial cell migration has an important role in angiogenesis [15], the effect of NELL-1 on wound closure was studied using a HUVEC scratch assay. We found that the PNG showed markedly induced HUVEC migration compared with the PG (Figure 2C).

Next, we performed a CAM assay to examine the angiogenic effects of NELL-1 in vivo. The assay results indicated that during embryonic vascularization, blood vessels in the NELL-1-treated group showed significantly greater vessel density and total numbers of segments and branching points compared with the control group. The results indicated that NELL-1 induced angiogenesis in vivo (Figure 3).

### 3.2. Pericytes and NELL-1 Promote Bone Formation In Vivo

To investigate the effectiveness of NELL-1 in vivo, we established an osteonecrosis model using mice and implanted femurs with pericytes alone (PG), or pericytes and NELL-1 (PNG) (Figure 4A). Plain radiographs and three-dimensional imaging indicated that femurs treated with pericytes, and NELL-1 (PNG) showed the greatest bone formation compared with other treatments (Figure 4B). The microCT results indicated that greater gains in BV/TV and BMD occurred in the PNG than in the PG (Figure 4C). Next, we examined the extent of osteocytic death via the quantification of empty lacunae in each group. We observed a synergistic decrease in the fraction of empty lacunae in PNG. In addition, Immunohistochemical analysis showed that CD31 and VEGF expression was higher in the PNG than in the PG, indicating that angiogenesis was induced in these groups (Figure 5).

### 3.3. NELL-1 Promotes the FGF2 Signaling Pathway

To identify the pathways associated with the NELL-1-dependent pro-angiogenic and osteogenic effects, we performed heatmap analysis using microarray data to compare the control versus NELL-1 treated pericytes. We identified four candidate genes (*FGF2*, *IL6*, *TGFB2*, and *VEGFA*) from among those upregulated by NELL-1 (Figure 6A,B), and also examined whether NELL-1 modulated the FGF2 pathway. NELL-1 induced an increase in FGF2 protein levels and the direct downstream targets of p-PLCγ, p-AKT, p-ERK, and p-eNOS. The results indicate that NELL-1 induced pro-angiogenic and osteogenic pathways (Figure 6C).

## 4. Discussion

The principal finding of this study relates to the NELL-1 dependent induction of angiogenesis and osteogenesis by pericytes. Pericytes combined with NELL-1 administration induced a synergistic effect during angiogenesis and osteogenesis [7], which are critically coupled processes for bone regeneration [34]. Vascular control is required for bone health. Blood vessels are key regulators of bone homeostasis because they serve as structural templates and provide nutrients and minerals [35]. Changes in vascular growth can negatively affect physiological bone healing and result in osteonecrosis [36]. The development of animal models to test novel cell-based therapeutic options that can prevent or delay osteonecrosis progression has been the focus of basic science and clinical research in this field. Cell-based therapies have been administered along with surgical treatment and resulted in significant improvements in survival compared with surgical treatment alone [6]. Given the growing evidence for the effectiveness and non-invasiveness of regenerative therapies, intensified efforts and studies to identify the efficacy of novel agents are necessary. To the best of our knowledge, this is the first study to use a mouse model with a necrotic bone fragment that mimics osteonecrosis in humans and highlights the potential of the combined application of stem cells (pericytes) and an osteoinductive protein (NELL-1) in vivo. We hypothesized that treatment with pericytes and NELL-1 could reverse osteonecrosis lesions. An osteogeneic and angiogenic potent effect was observed after local delivery, which was supported by radiographic images and histologic findings.

Large animals such as canines [37], pigs [38], and sheep [39] have been used as models to induce osteonecrosis. However, designing a series of experiments with these animals can be a limiting factor due to high costs. The use of rodents such as mice incurs relatively lower costs and is feasible for studies having large sample sizes. However, mouse models are somewhat small for surgical procedures. An ideal animal model should be cost-effective and large enough to perform various procedures. Considering these points, we selected the femoral shaft in mice instead of the proximal or distal end of the femur for analysis. We believe that our mouse model may provide the benefit of low cost and relatively large bone size to enable surgical procedures.

The regenerative potential of pericytes depends on the tissue of origin. We isolated and used AD pericytes that show efficient osteogenic differentiation [40]. To evaluate the effect of NELL-1 addition on pericytes, the concentration of NELL-1 required in the assay was analyzed and was based on the results of Zhang et al.’s study [25], which showed the effect of NELL-1 on bone formation. In addition, the effect of NELL-1 on pericyte proliferation using viability and WST-1 assays was assessed. The proliferation of pericytes without NELL-1 treatment and those treated with 100 ng/mL or 800 ng/mL NELL-1 was compared. Our results showed that no significant differences were observed between the three groups (Data not shown), indicating that the direct effect of NELL-1 on pericyte proliferation was not significant. We hypothesized that concentrations between 100 and 800 ng/mL were feasible for subsequent assays; and selected the highest concentration (800 ng/mL) to maximize the effect on proliferation and assessed osteogenic differentiation (Data not shown).

The effect of the pericyte-endothelial cell ratio on angiogenic potential has been examined [40]. Based on their results, Herrmann et al. used 10% pericytes and 90% HUVECs in their analyses [40]. We used a 1:20 ratio of pericytes:HUVECs that was chosen based on an optimal vascular tube growth and stability protocol (VascuNet™ Pericyte Co-Culture Assay; ESI BIO, Alameda, USA). We found synergistic effects between pericytes and NELL-1, and thus, the methods used in our analysis are feasible options for future studies.

Tube formation in HUVECs treated with NELL-1 without pericytes did not differ significantly compared with the control (Figure 2A), and NELL-1 did not have a direct effect on tube formation. Taken together, the results indicated that NELL-1 affected cell migration when used in combination with pericytes and affected the efficacy of pericytes due to an increase in FGF2 signaling. FGF2 has a well-characterized ability to regulate the growth and function of vascular cells such as endothelial and smooth muscle cells, and is a growth factor that increases vascularization, induces angiogenesis [41], and stimulates mitogenesis of mesenchymal progenitors and osteoblasts [42]. Future studies should aim at understanding the detailed mechanisms underlying NELL-1 activity when combined with pericytes, especially in the context of FGF2-dependent signaling.

This study has a few limitations. As the study shows the results of preclinical research, additional studies and evidence are required for clinical application. Nevertheless, we believe that our mouse model is relevant for assessing bone regeneration because treatment with pericytes and NELL-1 resulted in robust bone and vessel formation. Therefore, we believe that our mouse model may simplify the analysis of osteonecrosis in humans. Further, the reliability of mouse models to study osteonecrosis is questionable, and our mouse model may serve as a reliable option because ostectomy using the customized jig and the Gigli saw is much simpler than cauterization of vessels to directly induce ischemic osteonecrosis [43]. For consistency, all surgical procedures in this study were performed by the senior author, an experienced orthopedic surgeon. Lastly, the sequence of vessel maturation in angiogenesis was not evaluated. However, we assessed bone regeneration that resulted from the critically coupled processes of angiogenesis and osteogenesis.

## 5. Conclusions

The results of our study indicated that AD pericytes combined with NELL-1 treatment synergistically enhanced the sequence of angiogenesis. We used a mouse model of osteonecrosis and identified significant bone formation using radiographic images and histologic analysis. Thus, combination therapy using AD pericytes and NELL-1 may have potential as a novel treatment for osteonecrosis.

## Figures and Tables

**Figure 1 cells-10-02244-f001:**
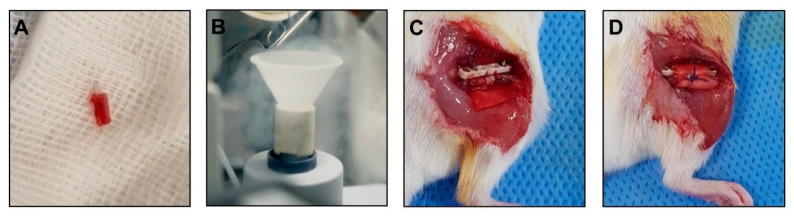
Procedures for the mouse model of osteonecrosis. (**A**) After ostectomy, bone fragments were removed. (**B**) The bone fragment was immersed in liquid nitrogen for 5 min. (**C**) An absorbable collagen sponge (ACS), with or without pericytes and NELL-1, was placed below the femoral shaft. Necrotic bone fragments were transferred to the site with defects. (**D**) The necrotic bone fragment was wrapped by suturing the ACS to prevent displacement.

**Figure 2 cells-10-02244-f002:**
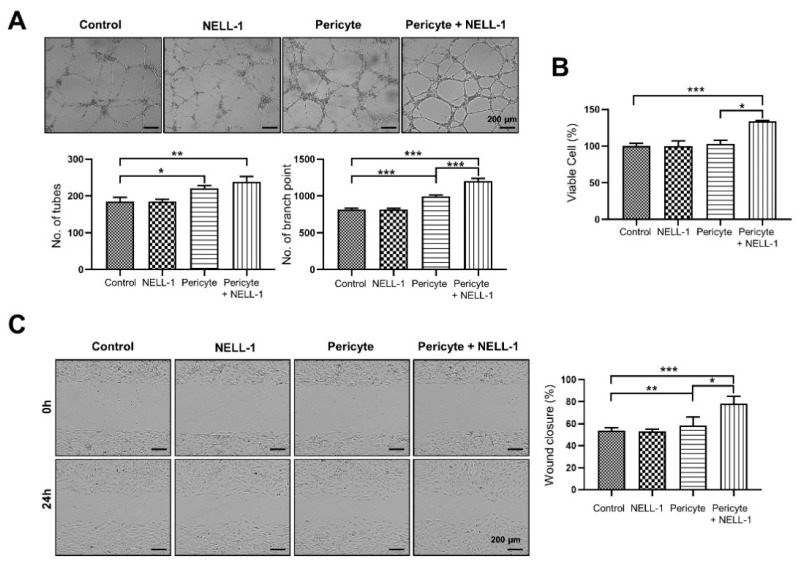
Pericytes and NELL-1 promote angiogenesis and migration of HUVECs. (**A**) Pericytes and NELL-1 promote tube formation of HUVECs. Matrigel-coated 96-well plates were seeded with HUVECs (4 × 10^4^) treated with pericytes and NELL-1 (800 ng/mL). After a 12 h incubation, a Nikon ECLIPSE Ti2 and analyzed using the Image J software program. Tube number was quantified using the Image-Pro Plus 6.0 software. (**B**) Pericytes and NELL-1 promote the viability of HUVECs in WST1 assays. 96-well plates were seeded with HUVECs (41.5 × 10^4^) treated with pericytes and NELL-1 (800 ng/mL). (**C**) Pericytes and NELL-1 promote migration of HUVECs in wound-healing assays. HUVECs were treated with or without NELL-1 (800 ng/mL) for 24 h. Cells were photographed using a JuLI™ Stage Cell History Recorder (NanoEntek, Guro-gu, Seoul, Korea). The migration rate was calculated using the JuLI™ Stage. Data were obtained from 3 independent experiments. Error bars represent mean ± SEM. * *p* < 0.05; ** *p* < 0.01; *** *p* < 0.001. Scale bar: 200 µm.

**Figure 3 cells-10-02244-f003:**
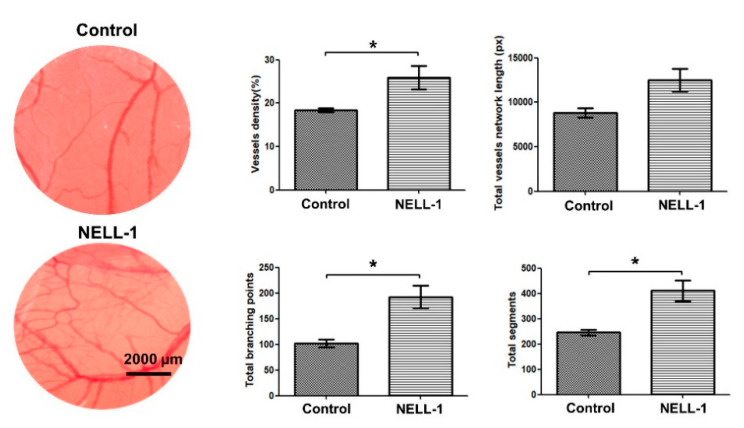
NELL-1 protein enhances angiogenesis. Representative images of chick chorioallantoic membrane in CAM assays. Fertilized eggs were treated with NELL-1 (800 ng/mL) for 72 h, along with non-treated controls. Microvessel images were captured using an EVOS FL Cell Imaging System (magnification: ×10). Data were obtained from 3 independent experiments. Error bars represent mean ± SEM. * *p* < 0.05. Scale bar: 2000 µm.

**Figure 4 cells-10-02244-f004:**
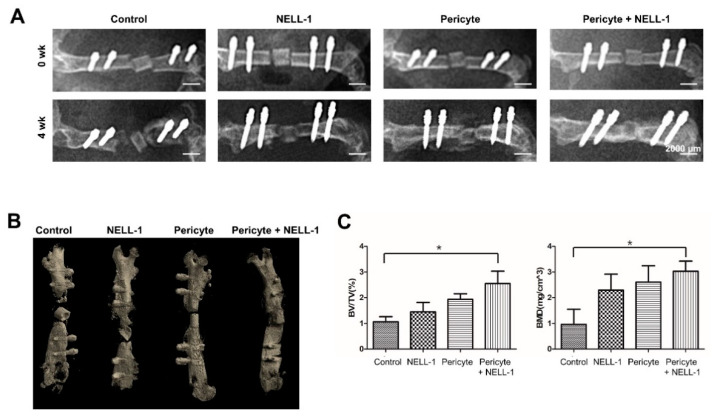
Pericytes and NELL-1 promote bone formation in an animal model for osteonecrosis. (**A**) The upper four images were taken immediately after surgical procedures. The lower four images were taken 4 weeks after surgical procedures and show the bone union of necrotic bone treated with pericytes and NELL-1. (**B**) Representative images of femurs isolated from pericytes treated with or without NELL-1 mice were generated using μCT. (**C**) BV/TV and BMD of the femurs (*n* = 6) were analyzed using a μCT scanner and CTAn software * Maximum effect compared with other treatments; * *p* < 0.05. Scale bar: 2000 µm.

**Figure 5 cells-10-02244-f005:**
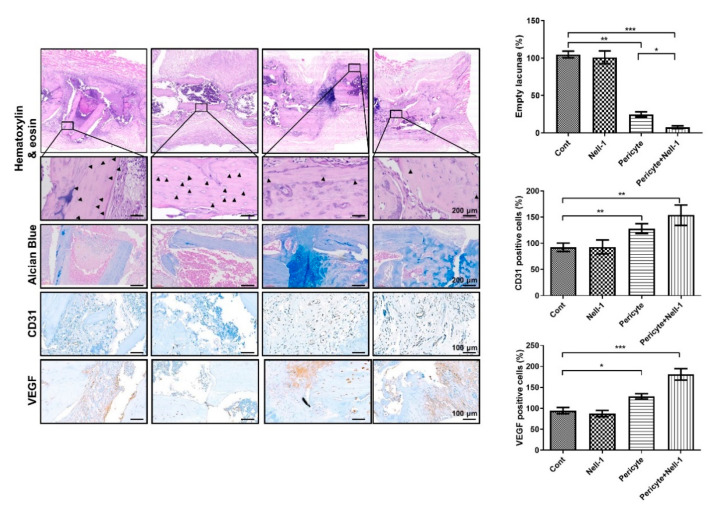
Immunohistochemical analysis of femur sections using hematoxylin and eosin (H&E) staining, Alcian Bluestain, and other relevant proteins. Immunohistochemical analysis of femur sections using hematoxylin and eosin (H&E) staining, Alcian Blue, and angiogenesis markers for CD31 and VEGF. At the end of the in vivo study, small femur sections taken from all groups were decalcified in EDTA, fixed in formalin, paraffin-embedded, sectioned, and immunostained using H&E (upper panel), Alcian Blue (second panel), CD31 (third panel), and VEGF (lower panel). The right panel shows the quantification data for immunohistochemical staining and the percentage of empty lacunae relative to the total lacunae. * *p* < 0.05; ** *p* < 0.01; *** *p* < 0.001. Images were taken at the magnification indicated. Scale bars indicate 200 μm for H&E, Alcian Blue images, and 100 μm for CD31, VEGF images.

**Figure 6 cells-10-02244-f006:**
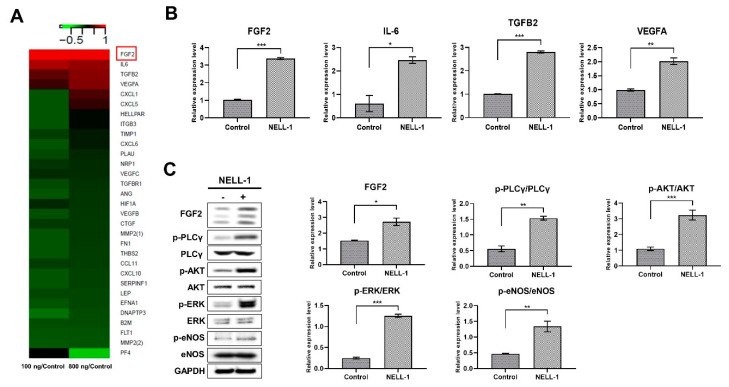
NELL-1 increases FGF2-AKT-eNOS pathway signaling. (**A**) Heatmap showing temporal expression patterns of angiogenesis-related mRNAs identified using pericytes treated with NELL-1 (100 ng/mL or 800 ng/mL). *Z*-scores of normalized read counts are indicated by the colored bars. Red: high expression; green: low expression. (**B**) Validation of selected differentially expressed mRNAs in pericytes treated with NELL-1 (800 ng/mL) via qRT-PCR. β-actin was used as a normalization control. Data were obtained from 3 independent experiments. Error bars represent mean ± SEM. (**C**) Western blot analysis of FGF2 pathway-related proteins after 72 h of stimulation with 800 ng/mL NELL-1. Data were obtained from 3 independent experiments. Error bars represent mean ± SEM. * *p* < 0.05; ** *p* < 0.01; *** *p* < 0.001.

**Table 1 cells-10-02244-t001:** Human adipose tissue samples used in the study.

Samples ^a^	Sex/Age	Past Medical History	SVF Yield	SVF Viability	Ratio of Pericytes in SVF	Pericytes Yield
1	M/75	Hypertension	22.4 × 10^6^	90.3%	0.27	5.46 × 10^6^
2	M/68	Diabetes mellitus	25.1 × 10^6^	85.2%	0.22	4.70 × 10^6^
3	F/73	None	19.0 × 10^6^	88.2%	0.24	4.02 × 10^6^
4	F/82	Hypercholesterolemia, Diabetes mellitus	31.4 × 10^6^	78.3%	0.31	7.62 × 10^6^
5	F/66	None	29.2 × 10^6^	79.2%	0.22	5.09 × 10^6^
6	F/72	None	22.3 × 10^6^	92.4%	0.19	3.91 × 10^6^
7	F/65	Hypertension, HBV carrier	15.1 × 10^6^	83.7%	0.29	3.67 × 10^6^
8	F/70	Hypertension	17.6 × 10^6^	87.9%	0.23	3.56 × 10^6^

^a^ Infrapatellar fat pads harvested during total knee arthroplasty were used for pericyte sorting.

## Data Availability

The authors confirm that the data supporting the findings of this study are available within the article.

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
