# Peer review of "Pro-Angiogenic and Osteogenic Effects of Adipose Tissue-Derived Pericytes Synergistically Enhanced by Nel-like Protein-1"

_cells, 2021, doi:10.3390/cells10092244_

Round 1

Reviewer 1 Report

The paper by An et al. “Pro-angiogenic and osteogenic effects of adipose tissue-derived pericytes synergistically enhanced by Nel-like protein-1” describes an interesting combined approach for the treatment of osteonecrosis. Results are interesting in the field of orthopedic surgery. However many issues need to be addressed before acceptance.

First, the authors claim to use adipose-derived pericytes isolated from the stromal vascular fraction. They are identified by cell sorting for their immunopositivity for CD146 and immunonegativity for CD34 and CD45.

Since pericytes lack a unique specific marker, this characterization appears rather weak. Moreover, the adipose stromal vascular fraction consists of a heterogeneous cell population, including mesenchymal stem cells, which are largely characterized by the same immune phenotype, and feature similar differentiation ability. The authors should address this issue and discuss whether some of these effects can be attributable to cells different from pericytes.

The site of adipose tissue harvesting is only mentioned in the table (Supplementary material)

Pericyte adipogenic differentiation, mentioned in “Methods”, is missing in the “Results”.

Pericyte viability test, described in “Methods”, is missing in the “Results” and reported only in the “Discussion”.

The Table and the figures reported in the supplementary material should be included and harmonized in the main manuscript, together with the existing ones.

Particular attention must be paid to abbreviations, which are not always explained. In some instances, they do not correspond to the legend. (Fig5: AB in figure, MT in the legend). Many scale bars are missing.

Antibody dilution should be reported.

The amount of HUVECs seeded in 96-well plates reported in “Methods” (40,000) does not correspond to the number reported in the legend to Figure 2 (15,000)

Many discrepancies are present in “Methods” and “Results” regarding the microscope and image software used. Please check.

The number of measurements used for statistical analysis is not specified.

Finally, the manuscript needs a substantial language revision.

Reviewer 2 Report

In the article entitled “Pro-angiogenic and osteogenic effects of adipose tissue-derived pericytes synergistically enhanced by Nel-like protein-1” by Hyun-Ju An and colleagues, the authors show that pericytes combined with NELL-1 synergistically enhanced angiogenesis and bone regeneration. Authors used a mouse model of osteonecrosis and found significant bone formation in radiographic images and histologic findings. The work makes a good impression. The animal model seems especially interesting.

However I have some minor questions and comments.

1) (line 52) Some words about current treatments and limitations of these techniques will make introduction more informative. The purpose of the work will clearer.

2) (line 57-58) What about paracrine regulation? Nowadays, secretome is considered as a major tool of mesenchymal stem cells (MSC) involvement in tissue repair and renewal in adults. According to Caplan MSC is population of pericytes. The direct cell contact is not main way. Do authors have alternative opinion? Moreover authors point to the key role of FGF2 in observed effects. FGF2 is a paracrine regulator.

3) (line 64-66) More information about NELL-1 is desirable. Why NELL-1, but not others? Can we use cocktail with 2 or more different cytokines for bone regeneration improvement?

4) Why did authors prefer pericytes, but not MSC? Why did authors prefer cells, but not extracellular vesicles?

(!) Therefore introduction must be improved.

5) This manuscript do not require supplementary. The data must be transferred from supplementary to main text.

6) (Supplementary, scheme 2) What is OM? Figures could be improved in order to help readers understand the data without time-consuming reference to the text. Undefined abbreviations should be avoided. Number of experiments (n) is unknown. The results of adipogenic differentiation were not shown, but this technique was described in Methods section.    

7) (Fig. 2) NELL-1 group must be present in the figure. It is important control.

8) (Fig. 3) Why did authors present NELL-1 only? The effects of pericytes, or pericytes conditioned medium, or NELL-1 activated pericytes is interesting too. Number of experiments (n) is unknown.

9) (Fig. 5) The images must be named. Number of experiments (n) is unknown.

10) (Fig. 6) Number of experiments (n) is unknown.

Author Response

Cells-130513

Title: Pro-angiogenic and osteogenic effects of adipose tissue-derived pericytes synergistically enhanced by Nel-like protein-1

Dear Reviewer

We are grateful to you and the reviewers for putting time and effort into reviewing our manuscript. The detailed and constructive comments provided valuable insights into our study. Having addressed the points, we would like to re-submit our Revised_manuscript. The changed or added contents & deleted contents were highlighted. Responses according to each comment are as follows.

[Comment 1]

In the article entitled “Pro-angiogenic and osteogenic effects of adipose tissue-derived pericytes synergistically enhanced by Nel-like protein-1” by Hyun-Ju An and colleagues, the authors show that pericytes combined with NELL-1 synergistically enhanced angiogenesis and bone regeneration. Authors used a mouse model of osteonecrosis and found significant bone formation in radiographic images and histologic findings. The work makes a good impression. The animal model seems especially interesting.

Response: First of all, we would like to appreciate your time and constructive comments on our manuscript. The detailed review provided valuable insights into our study.

[Comment 2]

However I have some minor questions and comments.

1) (line 52) Some words about current treatments and limitations of these techniques will make introduction more informative. The purpose of the work will clearer.

Response: Thank you for your kind comment. As you recommended, we have added the corresponding description. Please see the 2nd paragraph in the Introduction section.

[Comment 3]

2) (line 57-58) What about paracrine regulation? Nowadays, secretome is considered as a major tool of mesenchymal stem cells (MSC) involvement in tissue repair and renewal in adults. According to Caplan MSC is population of pericytes. The direct cell contact is not main way. Do authors have alternative opinion? Moreover authors point to the key role of FGF2 in observed effects. FGF2 is a paracrine regulator.

Response: Thank you for this valuable comment.

We fully agree with the reviewer’s opinion that FGF2 may play an essential role in paracrine signaling. We have performed mRNA sequencing to verify which gene expression is directly regulated by pericyte and Nell-1, resulting in the promotion of angiogenesis. The result shows that mRNA expression of FGF2 increased the most, which provides the insight that endogenous expression, as well as paracrine effect, are regulated altogether.

“Intrinsic FGF2 and FGF5 promotes angiogenesis of human aortic endothelial cells in 3D microfluidic angiogenesis system” confirmed that angiogenesis was regulated via direct suppression of FGF2 expression, which corresponds to our study. In addition, “FGF2 Translationally Induced by Hypoxia Is Involved in Negative and Positive Feedback Loops with HIF-1α” showed that FGF2 protein expression is up-regulated under hypoxia status. Based on these studies, it is thought that FGF2 plays an important role as an endogenous gene expression regulator as well as a paracrine regulator. We added sentences briefly describing this point in Discussion. Please see the 5th paragraph in the Discussion section

[Comment 4]

3) (line 64-66) More information about NELL-1 is desirable. Why NELL-1, but not others? Can we use cocktail with 2 or more different cytokines for bone regeneration improvement?

Response: Thank you for your valuable comment. As you recommended, we have added and modified the corresponding contents. As a regenerative therapeutic option in the field of orthopedic surgery, significant effects of human perivascular stem cells combined with NELL-1 have been demonstrated in the previous animal models. Considering the need for novel therapeutic options in ONFH, we focused on their synergistic effects and used a mouse model with a necrotic bone fragment. Please see the 3rd and 5th paragraphs in the Introduction section..

[Comment 5]

4) Why did authors prefer pericytes, but not MSC? Why did authors prefer cells, but not extracellular vesicles?

Response: Thank you for your kind comment. Especially, we have added the contents related to the advantages of pericytes. Please see the 4th paragraph in the Introduction section.

[Comment 6]

(!) Therefore introduction must be improved.

Response: As mentioned-above, we have revised the Introduction section based on your valuable comments.

[Comment 7]

This manuscript do not require supplementary. The data must be transferred from supplementary to main text.

Response: Thank you for your valuable comment. As you recommended, we have moved the contents to the main manuscript to make the statements more appropriate and convincing based on the findings of this study.

[Comment 8]

6) (Supplementary, scheme 2) What is OM? Figures could be improved in order to help readers understand the data without time-consuming reference to the text. Undefined abbreviations should be avoided. Number of experiments (n) is unknown. The results of adipogenic differentiation were not shown, but this technique was described in Methods section.

Response: Thank you for your valuable comment.

First, we apologize for the mistake with the abbreviations. It has now been corrected.

Second, we have added the number of experiments. Please see each Figure legend.

Finally, as you pointed out, we have deleted Method of adipogenic differentiation. By using FACS, pericytes were isolated and cultured. We confirmed their multi-potency of osteogenic and adipogenic differentiation. However, this data is currently not included in this paper. We apologize for the confusing descriptions.

[Comment 9]

7) (Fig. 2) NELL-1 group must be present in the figure. It is important control.

Response: Thank you for your valuable comment. Fig. 2 has now been modified accordingly.

[Comment 10]

8) (Fig. 3) Why did authors present NELL-1 only? The effects of pericytes, or pericytes conditioned medium, or NELL-1 activated pericytes is interesting too. Number of experiments (n) is unknown.

Response: Thank you for your valuable comment. Unlike HUVEC, chorioallantoic membrane has been reported to contain pericyte [1, 2]. we have added the number of experiments. Please see Figure 3 legend.

[Comment 11]

9) (Fig. 5) The images must be named. Number of experiments (n) is unknown.

Response: Thank you for your valuable comment. We have added the corresponding contents. Please see Figure 5 legend.

[Comment 12]

10) (Fig. 6) Number of experiments (n) is unknown.

Response: Thank you for your kind comment. we have added the corresponding number of experiments. Please see Figure 6 legend.

As you can see, we have performed extensive analysis and done our best to address each issue. We hope that these revisions have strengthened our manuscript to better meet the requirements of your prestigious journal.

Sincerely,

Soonchul Lee, M.D., Ph.D.

Department of Orthopaedic Surgery, CHA Bundang Medical Center, CHA University

59 Yatap-ro, Bundang-gu, Seongnam-si, Gyeonggi-do, Republic of Korea

Zip code: 13496

Tel. +82-31-780-5289, Fax.+ 82-31-708-3578

E-mail address: [email protected]

References

  1. Nico B, Ennas MG, Crivellato E, Frontino A, Mangieri D, De Giorgis M, et al. Desmin-positive pericytes in the chick embryo chorioallantoic membrane in response to fibroblast growth factor-2. Microvasc Res. 2004;68(1):13-9
  2. Kurz H, Fehr J, Nitschke R, Burkhardt H. Pericytes in the mature chorioallantoic membrane capillary plexus contain desmin and alpha-smooth muscle actin: relevance for non-sprouting angiogenesis. Histochem Cell Biol. 2008;130(5):1027-40.

Round 2

Reviewer 1 Report

An et al. “Pro-angiogenic and osteogenic effects of adipose tissue-derived pericytes synergistically enhanced by Nel-like protein-1”

After checking the revised manuscript, I noticed that some issues have been addressed but some incongruities still need to be clarified.

Figure 2: The “novel” panel A is missing in the figure. Former panels “A” and “B” should be renamed as “B” and “C”, according to the figure legend.

In my opinion, the “Supplemental Table A” should be included in the main manuscript as “Table 1” in the section “Methods”

Statistical analysis: The authors must decide if error bars represent SEM (in the text) or SD (in the legends).

Scale bars are mentioned in the legends but are still missing in the figures.

Author Response

Cells

RE: Cells-1305013

Dear Reviewer

Thank you very much for the response of August 11, 2021, regarding our manuscript entitled “Pro-angiogenic and osteogenic effects of adipose tissue-derived pericytes synergistically enhanced by Nel-like protein-1” together with the comments from the reviewer. According to your suggestions, we conducted additional revised our manuscript.

Our alterations after the reviewers’ comments are as follows: 

[Comment 1]

An et al. “Pro-angiogenic and osteogenic effects of adipose tissue-derived pericytes synergistically enhanced by Nel-like protein-1”

After checking the revised manuscript, I noticed that some issues have been addressed but some incongruities still need to be clarified.

Figure 2: The “novel” panel A is missing in the figure. Former panels “A” and “B” should be renamed as “B” and “C”, according to the figure legend.

Response: Thank you for your kind comment. We apologize for the mistake, and it has now been corrected.

[Comment 2]

In my opinion, the “Supplemental Table A” should be included in the main manuscript as “Table 1” in the section “Methods”

Response: Thank you for your valuable comment. As you recommended, we have included Table 1 in the section Methods. Please see the line: 91, 103.

[Comment 3]

Statistical analysis: The authors must decide if error bars represent SEM (in the text) or SD (in the legends).

Response: Thank you for your valuable comments. Statistical analysis in the “Methods” was clearly corrected. We apologize for the confusing descriptions.  

[Comment 4]

Scale bars are mentioned in the legends but are still missing in the figures.

Response: Thank you for your kind comments. We apologize for the mistake. We have corrected the corresponding contents. Scale bar has been included in all figures in the revised manuscript.

We are grateful that the manuscript has been improved satisfactorily and hope that it would be accepted for publication in the Cells. Thank you once again for your consideration.

Very sincerely yours, 

Soonchul Lee, M.D., Ph.D.

Department of Orthopaedic Surgery, CHA Bundang Medical Center, CHA University

59 Yatap-ro, Bundang-gu, Seongnam-si, Gyeonggi-do, Republic of Korea

Zip code: 13496

Tel. +82-31-780-5289, Fax.+ 82-31-708-3578

E-mail address: [email protected]